# Microsphere-Based Microsensor for Miniature Motors’ Vibration Measurement

**DOI:** 10.3390/s23229196

**Published:** 2023-11-15

**Authors:** Kaichuan Xu, Chunlei Jiang, Qilu Ban, Pan Dai, Yaqiang Fan, Shijie Yang, Yue Zhang, Jiacheng Wang, Yu Wang, Xiangfei Chen, Jie Zeng, Feng Wang

**Affiliations:** 1Key Laboratory of Intelligent Optical Sensing and Manipulation of the Ministry of Education, National Laboratory of Solid State Microstructures, Engineering Research Center of Precision Photonics Integration and System Application of the Ministry of Education, College of Engineering and Applied Sciences, Institute of Optical Communication Engineering, Nanjing University-Tongding Joint Lab for Large-Scale Photonic Integrated Circuits, Nanjing University, Nanjing 210023, China; xukaichuan_nepu@163.com (K.X.); pdai@nju.edu.cn (P.D.); 522023340013@smail.nju.edu.cn (J.W.); chenxf@nju.edu.cn (X.C.); 2College of Electrical and Information Engineering, Northeast Petroleum University, Daqing 163318, China; jiangchunlei_nepu@163.com; 3State Key Laboratory of Mechanics and Control for Aerospace Structures, Nanjing University of Aeronautics and Astronautics, No. 29 Yudao Street, Nanjing 210016, China; zj2007@nuaa.edu.cn

**Keywords:** microsensor, microsphere, photonic nanojet, miniature motor, vibration measurement

## Abstract

We present a microsphere-based microsensor that can measure the vibrations of the miniature motor shaft (MMS) in a small space. The microsensor is composed of a stretched fiber and a microsphere with a diameter of 5 μm. When a light source is incident on the microsphere surface, the microsphere induces the phenomenon of photonic nanojet (PNJ), which causes light to pass through the front. The PNJ’s full width at half maximum is narrow, surpassing the diffraction limit, enables precise focusing on the MMS surface, and enhances the scattered or reflected light emitted from the MMS surface. With two of the proposed microsensors, the axial and radial vibration of the MMS are measured simultaneously. The performance of the microsensor has been calibrated with a standard vibration source, demonstrating measurement errors of less than 1.5%. The microsensor is expected to be used in a confined space for the vibration measurement of miniature motors in industry.

## 1. Introduction

Miniature motors are widely used in various applications, such as aerospace, robotics, medical machinery, and intelligence manufacturing, mainly because of their high efficiency, high speed, fast response, good stability, and other superior characteristics [1,2,3,4,5]. Miniature motors have many parameters, among which the vibration of the miniature motor shaft (MMS) is one of the important parameters because the vibration of MMS is closely related to system control, condition monitoring, fault diagnosis, and model systems analysis. Therefore, a reliable technology for the vibration measurement of MMS is particularly important.

In the field of vibration measurement for motors, various types of sensors have been utilized in past works. These sensors include eddy current sensors, capacitive sensors, linear variable differential transformer (LVDT) displacement sensors, laser self-mixing interference (LSMI) sensors, digital holographic interference (DHI) sensors, etc.

Eddy current sensors are a relatively mature method for vibration detection, but in environments with strong electromagnetic interference, measurement errors are significant [6]. Capacitive sensors are widely used to measure physical quantities such as displacement and vibration, but there is the problem of edge effect, and the output signals are nonlinear, so they are inadequate for measuring vibrations with small amplitudes [7]. LVDT displacement sensors [8] are contact methods susceptible to issues such as wear and heat, making them unsuitable for vibration measurement in MMS. LSIM sensors have only one optical path and are compact. However, due to speckle interference, the measurement accuracy is low [9].

The DHI sensors are capable of measuring diverse vibration parts of a rotating shaft concurrently, but the sharpness of the interference fringes decreases sharply with increasing amplitude, and the measurement range is limited [10].

The above methods can be used to measure the vibration of the motor shaft in a specific environment. However, all of the above methods make it difficult to measure the axial and radial vibration of miniature motors, with diameters that are sub-millimeters or even smaller, in a confined space. In this paper, we propose a microsensor based on a microsphere for the vibrations measure of MMS. Compared with the currently available vibration sensors, the microsensor proposed by us is more compact in size and possesses a strong capability for light focusing and collection, which can simultaneously detect both axial and radial vibrations of miniature motors within a confined space. As far as we are aware, this represents the inaugural application of a microsphere-based microsensor for the purpose of measuring vibrations in miniature motors.

This study’s contributions can be outlined in the following manner:
(1)We combine a stretched single-mode optical fiber with a microsphere, with a diameter that is 5 μm, to fabricate a microsensor with a size in the micrometer scale.(2)The microsensor we propose overcomes difficulties encountered via conventional sensors in measuring the vibration of miniature motor shafts with diameters less than 1 mm or even microns within a confined space.(3)By exploiting the microsphere’s focusing properties (PNJ phenomenon), it becomes possible to accurately concentrate a sub-micron-sized light spot onto the surface of the measured rotating shaft. This, in turn, amplifies the reflected light intensity from the axial and radial positions of the miniature motor and enhances the signal-to-noise ratio of the measured signal, ultimately leading to improved measurement accuracy.(4)Only one light source is used to realize the measurement of the radial vibration and the axial vibration of the miniature motor simultaneously, and the measurement structure is compact.

## 2. Theory of PNJ and Fabry–Perot Interference

### 2.1. Principle of the PNJ Generated by a Dielectric Microsphere

When a wavelength-scale microsphere is exposed to a plane wave, the PNJ phenomenon occurs near its shadowed side. PNJ offers the benefits of high intensity and non-attenuation. Its focal spot is smaller than the diffraction limit. Thus, it is a potential method to enhance the sensing light for tiny objects. It is similar to the focal spot formed by a traditional lens, but their formation mechanism and properties are different. The Mie theory offers a comprehensive explanation of how light interacts with spherical particles of any size, allowing for an understanding of the PNJ phenomenon [11]. By applying Mie theory, it becomes evident how the field enhancement produced via dielectric spheres changes from a dipole structure in the Rayleigh scattering regime (where r ≪ λ) to a jet-like structure in mesoscale spheres (where r ∼ 1−30λ), with r representing the radius of the dielectric sphere and λ representing the wavelength of the plane wave [12].

The PNJ has many properties, which depend on the refractive index, the shape, the size of the microsphere, and the wavelength of the incident light [13]. In this study, the essential characteristics of the PNJ under consideration encompass the following parameters: full width at half maximum (FWHM), working distance (WD, defined as the distance from the tip end to the PNJ’s maximum intensity), peak light intensity, and intensity decay length (IDL) [14,15]. Due to the FWHM being narrower than the diffraction limit, it can be precisely directed towards the measurement position of the MMS, thereby reducing Doppler scattering interference to a minimum. The WD establishes the separation between the MMS measurement location and the surface of the microsphere. When the WD is small, the MMS may touch the microsphere and damage it. The IDL should be long enough, so the PNJ can always irradiate at the measured position of the MMS during the measurement process. The beam intensity of a PNJ can be increased hundreds of times compared to the incident light, thereby enhancing the intensity of scattered or reflected light from the MMS measurement location.

Taking the above parameters into account, we use COMSOL software6.0 to describe a simulation of a PNJ formed by a microsphere with a diameter of 5 μm. The refractive index of the microsphere (n_1_) is 1.59, and the refractive index of the background medium (n_2_) is 1. The incident light is a polarization-maintaining light source with a wavelength of 1550 nm and a power of 1.8 mW. The simulation diagram in Figure 1 illustrates the characteristics of the PNJ.

According to the simulation results depicted in Figure 1, the PNJ exhibits certain properties. The FWHM is roughly 690 nm, which symbolizes the width of the jet-like structure. The FWHM is located at approximately 16 μm along the axis, signifying the position where the PNJ’s maximum intensity is observed. The intensity decay length, approximately 20 μm, signifies how quickly the intensity of the PNJ decreases as you move away from its central axis. Furthermore, the PNJ center exhibits the highest beam intensity, indicating a concentrated energy distribution in that region. Hence, it has the capability to amplify the light intensity of the scattered or reflected light from the MMS’s measurement location to the optical fiber. In summary, the simulation shows that the PNJ formed by the given parameters satisfies the desired characteristics for measuring vibrations in MMS.

### 2.2. Phase Demodulation Principle of Fabry–Perot Interference Signal

The microsensor proposed in this paper is made of a stretched fiber and a microsphere. In recent years, fiber-optic sensors using Fabry–Perot (F-P) cavities as sensing structures have been widely used [16,17,18]. These sensors usually consist of two parallel surfaces in close proximity and use multi-beam interference as the sensing mechanism to measure external parameters [19,20]. In this study, the air end face of the microsphere and the radial position of the MMS together create an F-P cavity, with a cavity length denoted as d_1_. The air end face of another microsphere, in conjunction with the axial position of the MMS, establishes an F-P cavity, with the F-P cavity’s length designated as d_2_. The schematic diagram is shown in Figure 2. When the miniature motor is running, d_1_ and d_2_ change, and the phase difference between the reference light and the sensing light changes with the change in the cavity length (d_1_ and d_2_). The axial and radial vibrations are reconstructed by demodulating the phase difference between the reference light and the sensing light using the multiple Hilbert (MHT) algorithm.

The electric vector of the reference light and the sensing light In Figure 2 are as follows:(1)$$E_{r}\left(t\right)=A_{1}\cos[2\pi ft+\varphi_{r}],$$
(2)$$E_{s}\left(t\right)=A_{2}\cos[2\pi ft+\varphi_{s}(t)].$$

Here, *A*_1_ represents the amplitude of the reference light, and *A*_2_ represents the amplitude of the sensing light. $\varphi_{r}$ and $\varphi_{s}(t)$ denote the phases of the two beams, respectively. *f* is the frequency of the light. We utilize the identical phase demodulation technique as described in Ref. [21], and the phase equation for the interference signal I(t) can be expressed as follows:
(3)$$I(t)=I_{r}(t)+I_{s}(t)+2\sqrt{I_{r}(t)I_{s}(t)}\cos[\varphi_{s}(t)-\varphi_{r}],$$
where $I_{r}(t)$ and $I_{s}(t)$ denote the output intensity of the reference light and the sensing light, respectively. The phase difference of $\varphi_{s}(t)$ and $\varphi_{r}$ is related to the external cavity length between the air end face of the microsphere and the surface of the MMS, which can be written as follows:
(4)$$\phi (t)=\varphi_{s}(t)-\varphi_{r}=\frac{4\pi n}{\lambda }d(t),$$
(5)$$d(t)=\phi (t)\frac{\lambda }{4\pi n},$$
where *n* represents the refractive index of the medium (*n* = 1 for the air). *λ* denotes the wavelength of incident light. d(t) ($d_{1}(t)$ and $d_{2}(t)$) represents the displacement of the MMS, where $d_{1}(t)$ denotes the radial displacement, and $d_{2}(t)$ denotes the axial displacement.

In this paper, the MHT algorithm is used to demodulate the phase of the interference signal [22]. The expression of the Hilbert transform can be written as follows:(6)$$H[\cos(\phi (t))]=\frac{1}{\pi }\int_{-\infty }^{+\infty }\frac{\cos(\phi (t))}{t-\tau }d\tau .$$

The Hilbert transform can introduce a π/2 phase shift to the initial interferometric signal. Thus, tan[(ϕ(t)] can be obtained via MHT algorithm. Finally, ϕ(t) can be obtained by performing phase unwrapping. The block diagram of the MHT algorithm is shown in Figure 3. After being low-pass filtered (LPF), the cosine signal in the first half cycle is converted into a sinusoidal signal via Hilbert conversion (H[]). The sinusoidal signal is divided by the cosine signal of the second half cycle to obtain the tangent signal. Then, the tangent signal is transformed via arctangent to obtain ϕ(t).

## 3. Experiments and Results

### 3.1. The Calibration Experiment and Results

To check the precision and accuracy of the microsensor, the performance of the microsensor was tested using a standard oscillator-type piezoelectric (PZT, modelP753.1CD, Physik Instrumente, karlsruhe, Germany) [23] with a closed-loop control resolution of 0.1 nm. Figure 4 illustrates the experiment setup of this structure. We employed a PZT controller (similar to a signal generator) to generate a sinusoidal signal, which was then sent to the PZT, causing the PZT to vibrate in the form of a sinusoidal signal. The reference signal in Figure 5 refers to the signal sent from the PZT controller to the PZT.

The light source used in the experiment is a distributed feedback laser (DFB) model S3FC1550 manufactured by THORLABS (Newton, NJ, USA). It has an output wavelength of 1550 nm and an output power of 1.8 mW. The laser emitted from the DFB is split into two 50:50 beams after passing through the coupler. One beam from port 2 of the coupler enters the mircosensor1 after passing through port 1 and port 2 of the circulator_1, which is used to detect the displacement of the PZT. Then, the interference light enters the PD1 after passing through port 2 and port 3 of the circulator1. Similarly, the other light beam emitted from port 3 of the coupler can detect the displacement of the PZT. The PD1 and PD2 are simultaneously connected to a data acquisition card (NI, USB-4431, National Instruments, Austin, TX, USA). The interference signals collected using the PD1 and PD2 are converted into digital signals via the data acquisition card. Finally, the computer processes the signals via MHT algorithm to obtain the real displacement of PZT.

We carried out vibration measurements larger than half wavelength in the experimental process. Four sets of experiments are conducted. The PZT was set to vibrate sinusoidally at 7 Hz with amplitudes of 3.1 μm, 6.2 μm, 7 μm, and 8 μm, respectively. Figure 5 shows a set of measurement results of the microsensor when the vibrate amplitude is 6.2 μm. The original signal acquired using the microsensor is the blue curve in Figure 5a. In Figure 5b, the red curve is the PZT driving signal (reference signal), and the blue curve is the reconstructed signal obtained via the MHT algorithm. We can see that the two curves are nearly identical. Figure 5c shows the error between the reference signal and the reconstructed signal. The maximum error is less than 0.15 μm.

Five repetitions of each experiment were conducted. The measurement outcomes for microsensor1 are presented in Table 1, while those for microsensor2 are displayed in Table 2. Both Table 1 and Table 2 reveal that the relative errors for microsensor1 and microsensor2 are both below 1.5%.

In order to verify the linearity of the vibration form of PZT and the PZT driving signal (reference signal), we utilized a PDV_100 (Polytec, Carlsbad, Germany) laser vibrometer to measure the PZT’s vibrations. Subsequently, signal processing was performed. The experiment setup is depicted in Figure 6. We set the PZT frequency to 5 Hz, with amplitudes of 2 μm, 4 μm, 6 μm, 8 μm, and 10 μm, respectively. The experiment results are shown in Table 3. The experiment results show that the errors between the PZT amplitude measured by the PDV_100 laser vibrometer and the PZT driving signal are small, and there is a strong linear relationship between the vibration mode of the PZT and its driving signal. There are two main sources of error: On the one hand, it is the measurement error of the PDV_100 laser vibrometer itself. On the other hand, it comes from the error introduced in the signal processing process.

### 3.2. Experiments and Results of the MMS Vibration Measurement

We conducted experiments to verify the feasibility of the proposed microsensor to measure the vibration of the MMS. Figure 7 shows the schematic diagram, and Figure 8 shows the device diagram. In Figure 8, in order to ensure the stability of the microsensor, we used super glue to fix the microsensor to a 5-axis stage, and the miniature motor was fixed to another 5-axis stage. By controlling two 5-axis stages, the relative position of the microsensor and the MMS can be controlled. And the two 5-axis stages are fixed on the two optical platforms, respectively, to prevent the stability of the microsensor from being affected when the miniature motor is running. We measure the axial and radial vibrations of the MMS of a hollow-cup gear motor 8520 (C.L.K, Shenzhen, China) [24] via two microsensors simultaneously. The rated voltage of the motor is 3.7 V (DC), the no-load current is 450 mA, the no-load speed is 48,000 r/min, and the diameter of the MMS is 0.6 mm. The measurement of vibrations at different positions along the MMS is of significant importance. Measuring vibrations at the front end of the MMS can help detect issues such as rotor imbalance and eccentricity. Measuring the middle end of the MMS can assist in identifying problems like rotor bending and misalignment while measuring at the back end of the MMS can aid in detecting bearing wear and rotor friction issues. So, we have tried to measure different parts of the radial position of the MMS. We measured the vibration of the front, middle, and back end of the MMS, respectively, and reconstructed signals.

In the experiment, electromagnetic interference will affect the detection of signals. We ground all equipment to minimize electromagnetic interference. In order to reduce Doppler speckle interference, we cover the protective layer of steel pipe around the microsensor, so the sensing light could be vertically irradiated to the axial and radial surfaces of the MMS.

The results obtained with the two microsensors are shown in Figure 9 and Figure 10, respectively. Figure 9a represents the original signal of the axial vibration, and Figure 9b represents the reconstructed signal of the axial vibration. Figure 10a,c,e represent the original signals of the front, middle, and back end of the MMS measured along the radial position, respectively. Figure 10b,d,f show the reconstructed signals of the front, middle, and back end of the MMS measured along the radial position, respectively. The above signals are detected when the miniature motor works under the rated voltage. The experiment results show that the proposed microsensor can measure the axial and radial vibration of the MMS well.

### 3.3. Experiments and Results of Sensitivity Measurement

The sensitivity of the microsensor that we proposed is the crucial technical parameter for measuring small vibrations. In MMS vibration monitoring, small vibration amplitudes may carry important information about motor performance and mechanical operation. So, we used a standard vibration source PZT to measure the sensitivity of the proposed microsensor. The experiment device is shown in Figure 11. The PZT was driven to perform vibrations with amplitudes of 10 to 100 nm, a step of 10 nm, and a frequency of 5 Hz. We used the microsensor to measure the vibration of the PZT. The relationship between the output voltage signal and the vibration amplitude is plotted, as shown in Figure 12. The abscissa represents the vibration amplitude of the PZT (input signal), and the ordinate represents the peak-to-peak voltage (output signal) in Figure 12. We perform curve fitting on the input–output data, and the slope of the curve fitting is the sensitivity. It can be seen from the experiment data that the sensitivity of the proposed microsensor is 0.7 mV/nm, which has a high sensitivity and can meet the requirements of MMS vibration measurement. When measuring MMS vibration using the microsensor we proposed, it is possible to monitor and detect potential problems in real time, such as unbalance, bearing wear, or other mechanical failures of the MMS.

### 3.4. The Comparative Experiment and Results

To further verify that the microsphere has good focusing performance and can enhance the reflected light, we measure the axial vibration and the radial vibration at the middle end of the same MMS using two stretched fibers without microspheres under the same conditions. The experiment structure diagram is shown in Figure 13, which is similar to that in Figure 7.

The experiment results of microsensor1 measuring the axial vibration are shown in Figure 14, and the experiment results of microsensor2 measuring the radial vibration are shown in Figure 15. As seen in Figure 14 and Figure 15, both the axial and radial vibration signals are degraded. On the one hand, the amplitude of the radial vibration signal and the axial vibration signal decreased by an order of magnitude. On the other hand, the axial vibration signal becomes unstable, and the signal-to-noise ratio of the radial vibration signal is reduced, which is because the light spot is too large to treat the reflecting surface of the tiny axis as a plane. So, the reconstructed signal has severe distortion. Therefore, we believe that microspheres play an important role in the measurement process.

## 4. Discussion

In addition to sensitivity, the microsensor has some other very important technical parameters, such as dynamic range, frequency response, etc. We have listed the important technical parameters of the microsensor in Table 4. In MMS vibration detection, vibration may occur in different frequency ranges. Frequency response indicates the microsensor’s ability to measure vibrations at different frequencies. Understanding the frequency response is critical because we need to ensure that the microsensor can cover the range of vibration frequencies that the miniature motor may experience. This will help us capture a variety of vibration events, from high-frequency running vibrations to low-frequency fault vibrations. The frequency response of the proposed microsensor is 0.5 Hz~100 kHz, which can meet the demand for MMS vibration detection. Due to different operating conditions, the vibration amplitude of the MMS may be different. The dynamic range of the proposed microsensor determines the range of vibration amplitude it can measure. The wide dynamic range enables the microsensor to handle different cases of vibration from small to large without saturating or distorting the signal. This is very important for monitoring the vibration of the MMS under different loads and speed conditions. The dynamic range of the microsensor is 10 nm~500 μm, and the wide dynamic range enables the microsensor to measure the vibration of the MMS.

When measuring the vibration of the MMS, the proper preparation of the MMS surface can improve measurement accuracy and repeatability. The following are some common surface preparation methods:
(1)Clean the Surface: Ensure that the MMS surface is clean and free from dust, grease, or other contaminants. Using a cleaner or solvent can help remove any residues that might interfere with measurements.(2)Smooth the Surface: Inspect the MMS surface for smoothness, ensuring there are no dents, protrusions, or irregularities. Irregular surfaces can introduce measurement errors.(3)Rust Removal: If there is rust on the shaft surface, it should be removed to ensure accurate measurements. Sandpaper, a wire brush, or other tools can be used to eliminate rust.(4)Surface Treatment: For high-precision measurements, consider surface treatment such as grinding or polishing to ensure the surface is smooth and uniform. Surface condition is crucial for accurate vibration measurements.

## 5. Conclusions

This study proposes a microsensor based on a microsphere for the vibration measurement of the MMS. The microsensor comprises a stretched fiber and a 5 μm diameter microsphere, with the air end face of the microsphere forming an F-P cavity with the measured position of the MMS. The PNJ phenomenon generated by the microsphere plays an important role in focusing the light on the tiny target and enhancing the signal intensity. The error of the microsensor is demonstrated to be less than 1.5% by using a standard vibration source PZT for calibration. We also performed comparative experiments for measuring the axial vibration and the radial vibration of the MMS using microsensors with and without microspheres. The sensing results demonstrate that the proposed microsphere-based microsensor has a significant enhancement to the sensing performance. The microsensor has the advantages of miniaturization, easy integration, and low cost, and can be easily fabricated in smaller sizes via micromachining, which will make the microsensor competent for sensing in tighter spaces. In particular, it can be used for vibration detection of micro-targets.

## Figures and Tables

**Figure 1 sensors-23-09196-f001:**
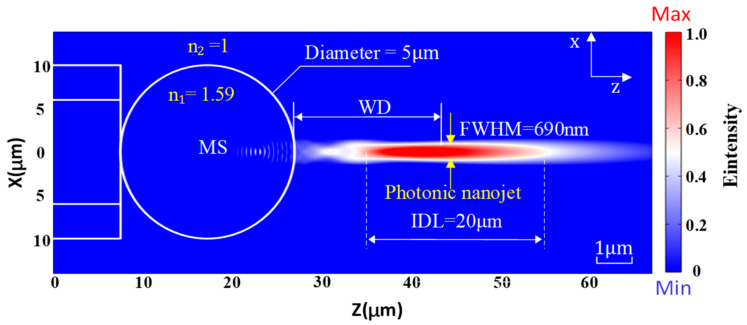
Simulation diagram of the PNJ. n_1_ is the refractive index of the MS. n_2_ is the refractive index of the background medium. WD represents the distance from the tip end to the PNJ’s maximum intensity. IDL represents the intensity decay length. FWHM is the full width at half maximum of the PNJ.

**Figure 2 sensors-23-09196-f002:**
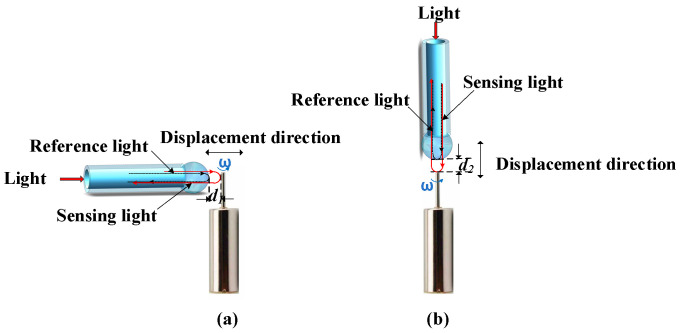
Schematic diagram of the two microsensors measuring the vibration of the MMS: (**a**) measuring the radial vibration; (**b**) measuring the axial vibration.

**Figure 3 sensors-23-09196-f003:**
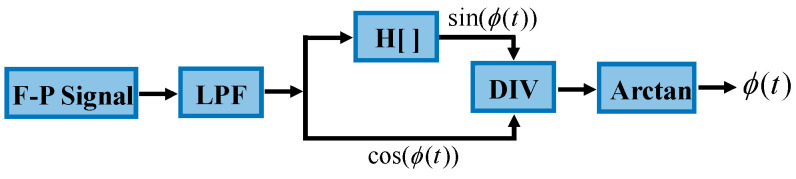
Block diagram of MHT algorithm.

**Figure 4 sensors-23-09196-f004:**
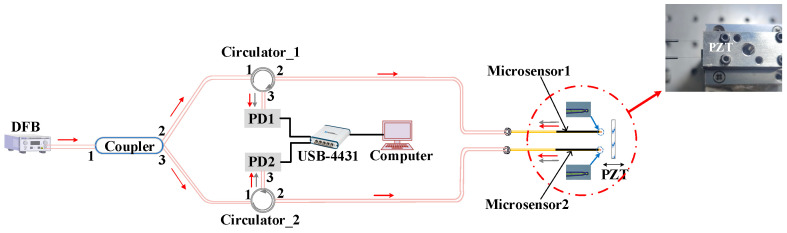
Schematic diagram of the PZT calibration experiment.

**Figure 5 sensors-23-09196-f005:**
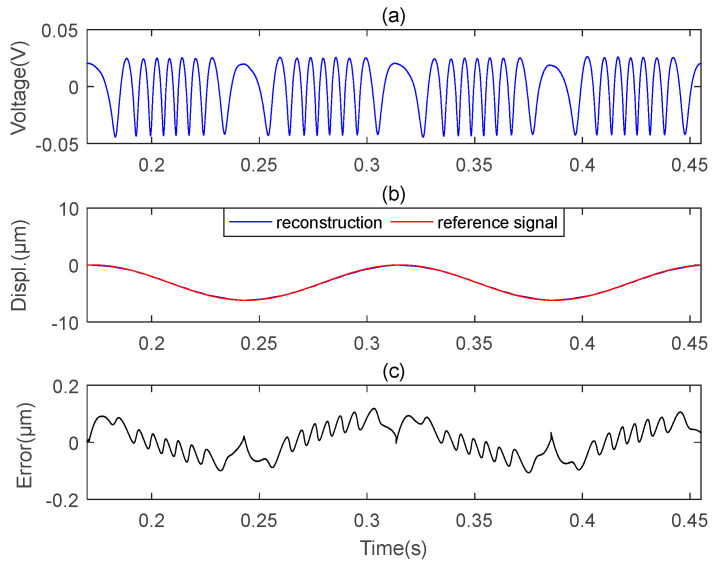
Experimental results of microsensor1. (**a**) Original F-P signal; (**b**) Reference signal and reconstructed signal; (**c**) Absolute error.

**Figure 6 sensors-23-09196-f006:**
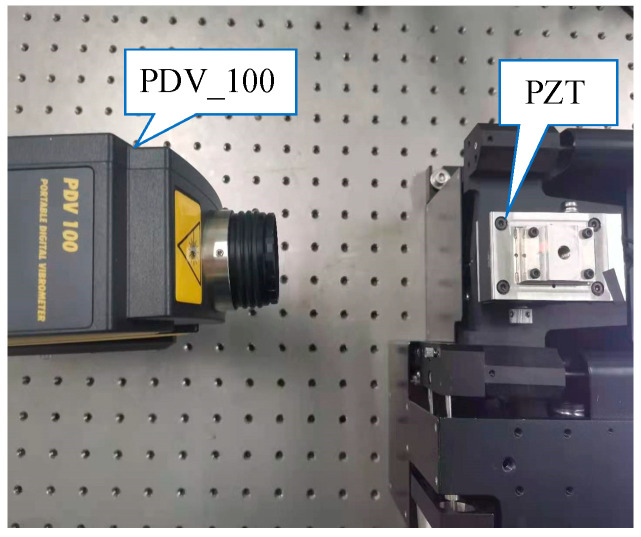
The experiment setup for validating the linear relationship between the PZT drive signal and the PZT vibration mode using the PDV_100 laser vibrometer.

**Figure 7 sensors-23-09196-f007:**
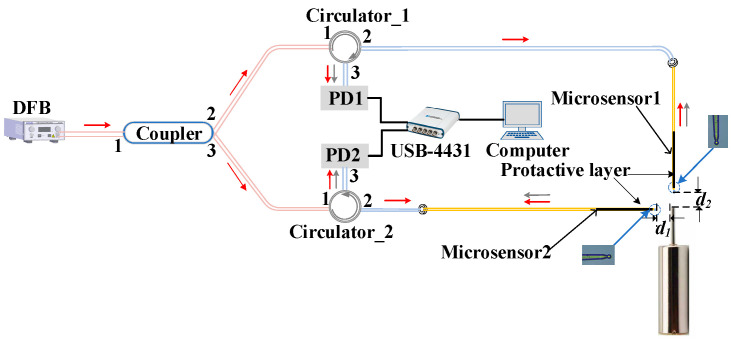
Schematic diagram of the vibration measurement of the MMS using microsphere-based microsensors.

**Figure 8 sensors-23-09196-f008:**
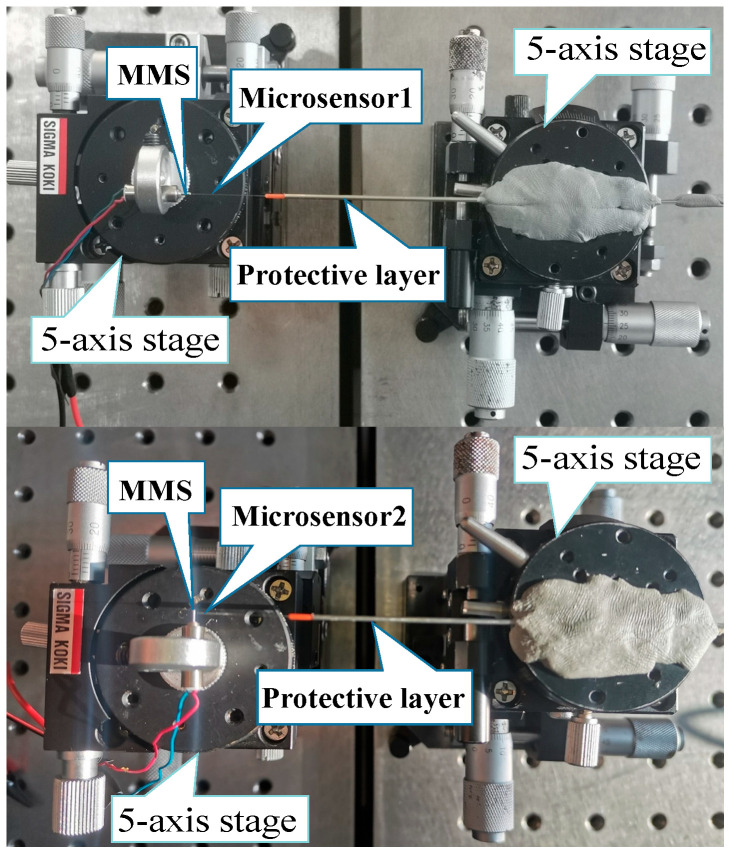
Setup diagram of the vibration measurement of the MMS using microsphere-based microsensors.

**Figure 9 sensors-23-09196-f009:**
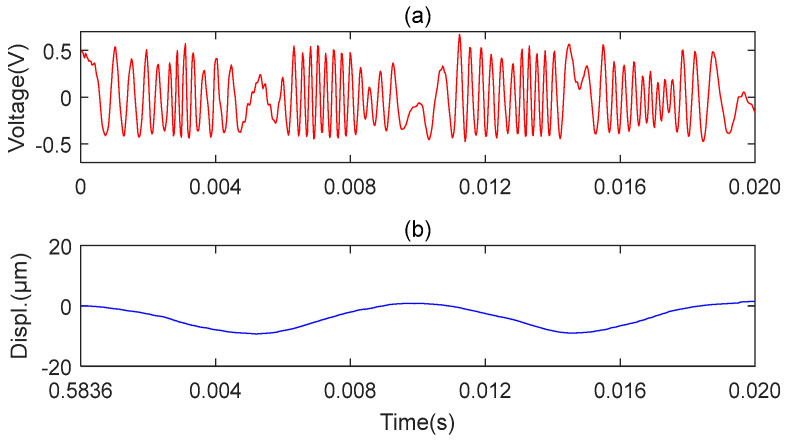
Microsensor1 measures the axial vibration. (**a**) Original axial vibration signal; (**b**) Reconstructed axial vibration via MHT algorithm.

**Figure 10 sensors-23-09196-f010:**
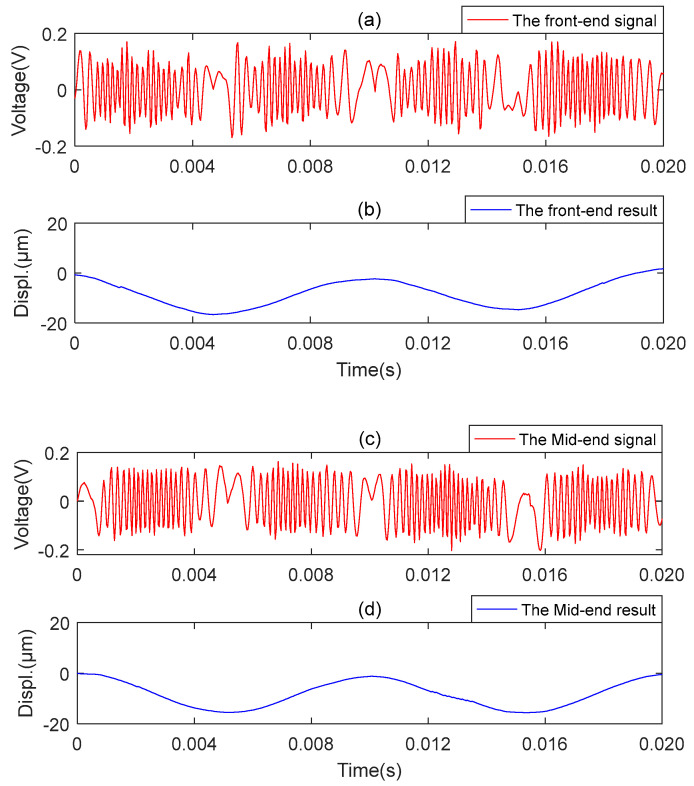
Microsensor2 measures vibrations at different locations in the radial direction. (**a**) The front-end original vibration signal; (**b**) The front-end measurement result. (**c**) The mid-end original vibration signal; (**d**) The mid-end measurement result. (**e**) The back-end original vibration signal; (**f**) The back-end measurement result.

**Figure 11 sensors-23-09196-f011:**
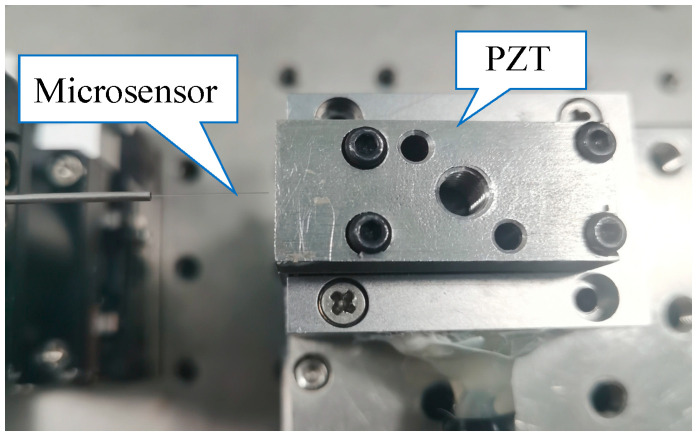
Experimental device for measuring the sensitivity of the microsensor using PZT.

**Figure 12 sensors-23-09196-f012:**
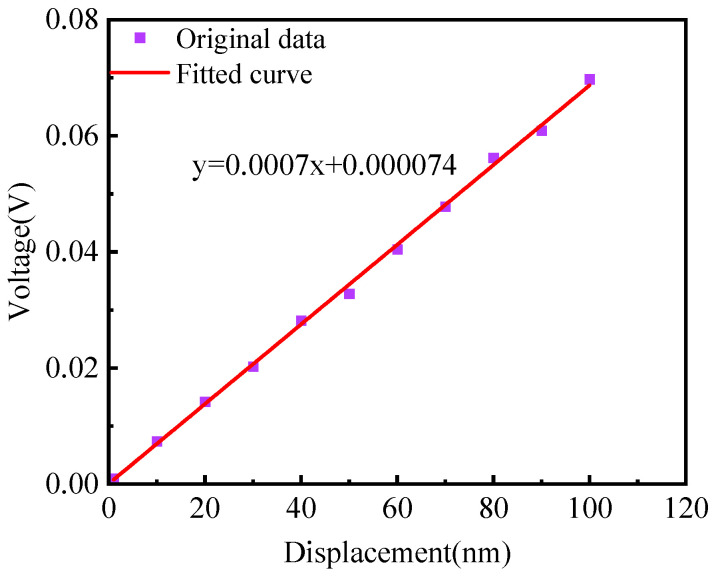
Experimental results for measuring the sensitivity of the microsensor.

**Figure 13 sensors-23-09196-f013:**
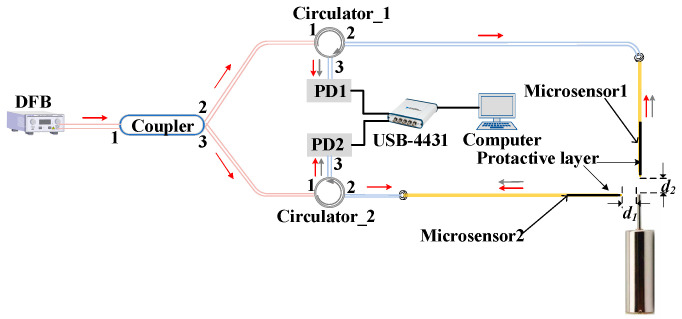
Schematic diagram of the vibration measurement of the MMS using two stretched single-mode fibers without microspheres.

**Figure 14 sensors-23-09196-f014:**
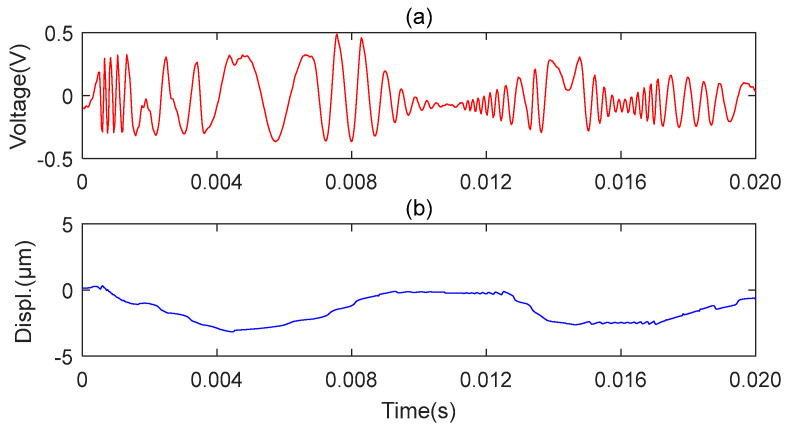
The result of the axial vibration measurement of the MMS using a stretched fiber without microsphere. (**a**) Original axial vibration signal; (**b**) Reconstructed axial vibration via MHT algorithm.

**Figure 15 sensors-23-09196-f015:**
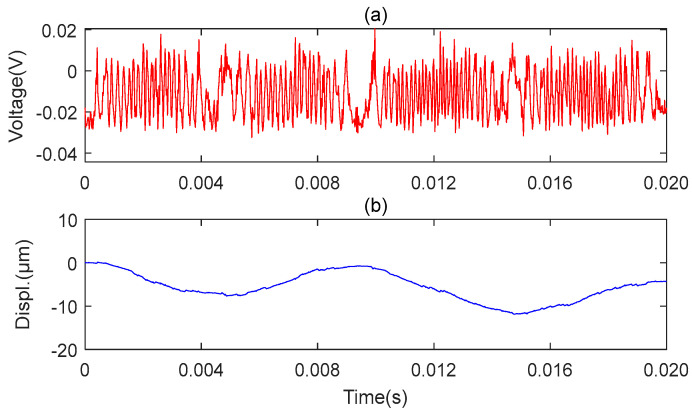
The result of the radial vibration measurement at the middle end of the MMS using a stretched fiber without a microsphere. (**a**) Original radial vibration signal; (**b**) Reconstructed radial vibration via MHT algorithm.

**Table 1 sensors-23-09196-t001:** Measurement results of the microsensor1.

Amplitude (μm)	Maximum Absolute Error (nm)	Average Absolute Error (nm)	Maximum Relative Error (%)
3.1	75	68	1.21%
6.2	99	90	0.80%
7	132	123	0.94%
8	155	138	0.97%

**Table 2 sensors-23-09196-t002:** Measurement results of the microsensor2.

Amplitude (μm)	Maximum Absolute Error (nm)	Average Absolute Error (nm)	Maximum Relative Error (%)
3.1	78	69	1.26%
6.2	95	88	0.77%
7	144	132	1.01%
8	165	156	1.03%

**Table 3 sensors-23-09196-t003:** Measurement results for validating the linear relation between the PZT drive signal and the PZT vibration mode.

Frequency (Hz)	PZT Amplitude (μm)	PDV_100 Amplitude (μm)	Maximum Absolute Error (nm)
5	2	1.970	30
5	4	3.957	43
5	6	5.960	40
5	8	7.953	47
5	10	9.956	44

**Table 4 sensors-23-09196-t004:** Technical parameters of the microsensor.

Parameter	Numerical Values
Resolution ratio	1 nm
Linearity	±1.5%
Working temperature	−40~100 °C
Sensing distance	<1 cm
Frequency response	0.5 Hz~100 kHz
Dynamic range	10 nm~500 μm
Relative error	<2%
Sensitivity	0.7 mV/nm
Repeatability	±1%

## Data Availability

Data are contained within the article.

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
