# Peer review of "Microsphere-Based Microsensor for Miniature Motors’ Vibration Measurement"

_sensors, 2023, doi:10.3390/s23229196_

Round 1
Reviewer 1 Report
Comments and Suggestions for Authors
In this manuscript the authors present a microsphere-based microsensor which can measure vibrations. The microsensor is composed of a stretched fiber and a microsphere with a diameter of 5 μm. I have several questions:
1. Figure 1 is not clear. The size of the sensor is not marked. The annotation is not clear.
2. As a vibration sensor, what’s the frequency response of the sensor?
3. The sensitivity and the dynamic range of the sensor should be given and discussed according to the test results.
Comments on the Quality of English LanguageModerate editing of English language required.
Reviewer 2 Report
Comments and Suggestions for Authors
The microsensor presented in this article is is based on a microsphere for vibrations measurement of
miniature motor shaft. This type of sensor could be helpful for the robotic technology as well as for many other technical areas. The microsensor comprises a stretched fiber and a 5 μm diameter microsphere, with the air end face of the microsphere forming an Fabry-Perot cavity with the measured position of the miniature motor shaft. The photonic nanojet phenomenon generated by the microsphere plays an important role for focusing the light on the tiny target and thus for enhancing the signal intensity. The error of the microsensor is demonstrated to be less than 1.5% by using a standard vibration source for calibration.
The authors performed comparative experiments for measuring the axial vibration and the radial vibration of the miniature motor shaft by using the microsensors with and without the microspheres. The sensing result demonstrate that the proposed microsphere-based microsensors have significant enhancement to the sensing performance. The microsensor has the advantages of miniaturization, easy integration and low cost, and can be easily fabricated in smaller sizes through micromachining, which will make the microsensor competent for sensing in tighter spaces.
The article is well written, the experiments well documented, references well selected, I would recommend to be published as it is .
Reviewer 3 Report
Comments and Suggestions for Authors
Dear Authors, thank you for teh interesting work.
I have few questions I would like you to address in a reviewed version of the paper.
You measured the vibration of a MMS, in a real application there will be kind of load /operator connedted to the MM and accessibility will be much lower. How will you cope with the limitations?
It would be interesting to have somewhere a tabele with technical specs of the sensors.. like frequency span, sensing distances, surface needs..
You are measuring "displacements" so the relative position of the sensor with respect of the moving object is very important. How do you assure the stability of the fibre/sphere set up? the pictures show a rather long beam at the end ov which the sphere is sitting..
You measured the vibration of a PZT and comparred the reference signal and the reconstructed response. What exactly is the reference signal? what is sent to the PZT? sure this has a linear relation with the vibration?
Could you justify why there is a peak in the error around 7nm in table 1 and 2?
Have you tried to measure on different shaft position? with which results?
How the surface could be treated to imporve the measurements?
Thanks for the effort
Comments on the Quality of English LanguageOverall english is acceptable.
